# Chemical and Functional Characterization of Propolis Collected from Different Areas of South Italy

**DOI:** 10.3390/foods12183481

**Published:** 2023-09-19

**Authors:** Giulia Grassi, Giambattista Capasso, Emilio Gambacorta, Anna Maria Perna

**Affiliations:** 1Department of Agricultural, Environmental and Food Sciences, University of Molise, Via De Sanctis 1, 86100 Campobasso, Italy; 2School of Agricultural, Forestry, Food and Environmental Sciences, University of Basilicata, Viale dell’Ateneo Lucano 10, 85100 Potenza, Italy; giambattista.capasso@unibas.it (G.C.); emilio.gambacorta@unibas.it (E.G.); anna.perna@unibas.it (A.M.P.)

**Keywords:** propolis, chemical and functional composition, antioxidant activity, Basilicata

## Abstract

This study investigated the chemical and functional characterization of propolis collected in southern Italy, in particular in Basilicata, a region rich in ecological and vegetative biodiversity. Sixteen samples of propolis, collected within a radius of 40 km from each other in the Basilicata region, showed significant differences between the chemical and functional parameters investigated: color index (L*, a*, b*; *p* < 0.05) and variation in chemical composition and antioxidant activities by ABTS and FRAP assays. In general, Lucanian propolis had a low content of waxes (*p* < 0.05) and a high content of resin (*p* < 0.05) and balsams (*p* < 0.05). The content of the total phenolic compounds and flavonoids was highly variable, as was the biological capacity. In conclusion, Lucanian propolis showed remarkable variability, highlighting significant diversification according to the geographical position and the diversity of the flora surrounding the apiary that the bees use as a source of resin. This study, therefore, contributes to the enhancement of the quality of propolis, laying the foundations for the production and marketing of propolis not only in the food industry but also in the pharmaceutical and cosmetic industries.

## 1. Introduction

Propolis is a natural product that is processed by bees with the addition of other substances such as wax, pollen and glandular secretions. It appears as a resinous substance that arises from the industrious work of foraging bees that feed on substances present on the bark and exudates of numerous plants [1,2].

Propolis is a product that is used inside the hive to protect and build borders and entrances for bees but also as an insulator thanks to its ability to be sticky in hot periods and rigid in cold periods. It is commonly called “bees glue” and it is an excellent biological defense against the proliferation of microorganisms [3,4]. Furthermore, bees use propolis as a natural remedy to prevent the decomposition of the carcass of other insects that settle in the hives and it is able to stabilize the internal temperature of the hive at around 35/37 °C. Finally, being a lipophilic substance, it prevents the penetration of water into the hive with a consequent stabilization of humidity and regulates the airflow in the hive [2]. In nature, there are numerous types of propolis that differ from each other in chemical composition, physical characteristics, color and other characteristics that make them unique. In fact, some researchers have noticed differences between various propolis samples in consistency; some were brittle and hard, while others were elastic and gummy. In general, propolis is characterized by 50% resin (phenolic compounds) and vegetable balms, 30% wax, 10% essential and aromatic oils, 5% pollen and 5% other substances, including also organic debris [5]. In reality, there are a number of factors that influence the composition of raw propolis: geographical area, botanical origin, seasonality, climatic temperatures and others. Also, the color of propolis is influenced by the geographical area and the plant source on which the bees feed [5]. The components that characterize this product are numerous: steroids, amino acids, phenols, terpenes, flavonoids, carbohydrates, aliphatic and aromatic acids and esters. Its use in cosmetic, pharmaceutical and food preparations is also increasing, thanks to the therapeutic, preventive and improvement activities of our body [6]. The biological effects of this matrix are a broad spectrum: antibacterial and antioxidant, antitumor, cardioprotective, antiviral, immunomodulatory, hepatoprotective, neuroprotective, antidiabetic, anti-inflammatory, anesthetic and antiallergic due to the abundant mineral content [7,8,9]. The need to chemically type propolis could be useful in order to officially include it within the health system, so as to guarantee the safety and quality of propolis in health and therapeutic fields. The chemical composition of propolis depends on the geographical and climatic characteristics of the place of collection, which makes propolis an extremely variable matrix and, consequently, makes it difficult to standardize the characteristics of propolis. Therefore, it is necessary to investigate its chemical composition and also the relative biological properties in order to contribute to enriching the partially already existing knowledge. In support, Graikou et al. [10] have demonstrated the presence of different propolis in the Mediterranean area in relation to its geographical origin. Gardini et al. [11] highlighted the need to investigate the variability of the composition of propolis collected in the Italian territory, according to the ecoregion of origin suggested by Blasi et al. [12]. Our study is inserted within this context and aims to quantify the physicochemical parameters, color index, total phenolic content and flavonoids and evaluate the antioxidant activity. In addition, the correlations between the parameters analyzed in propolis of different geographical origins of southern Italy were also studied, with particular reference to the Basilicata region, a region rich in ecological and vegetative biodiversity.

## 2. Materials and Methods

### 2.1. Chemicals and Apparatus

The 2,2′-azino-bis-(3-ethylbenzothiazoline-6-sulfonic acid) (ABTS), 2,4,6-tripyridyl-s-triazine (TPTZ), potassium persulfate, hydrochloric acid, ferric chloride, iron(II) sulfate heptahydrate, sodium phosphate, sodium hydroxide and ammonium persulfate were purchased from Sigma-Aldrich (Milan, Italy). The phenolic compounds, gallic acid and quercetin, were purchased from Sigma Chemical Co. (St Louis, MO, USA). Analytical grade reagents, such as sodium carbonate, potassium hydroxide, Folin-Ciocalteu reagent, ethanol, methanol and hexane were obtained from Panreac (Barcelona, Spain). Aluminum chloride, potassium ferricyanide, ferric chloride and trichloroacetic acid were from Sigma Chemical Co. (St. Louis, MO, USA). The water was treated in a Milli-Q water purification system (TGI Pure Water Systems, Brea, CA, USA). Coomassie Brilliant blue G250 was purchased from Bio-Rad (Richmond, CA, USA). The spectrophotometer UV-VIS Spectrophotometer 1204 (Shimadzu, Japan) was used. MINOLTA Chromameter CR-300 (Minolta Camera Corp., Meter Division, Ramsey, NJ, USA) equipped with a D65 illuminant, 10° Observer and zero and white calibration was used to measure the color parameters (CIE L*, a*, b*).

### 2.2. Propolis Samples

A total number of 16 propolis samples were taken directly from hives located in 8 different areas of the Basilicata region (Italy), according to availability and beekeeping activity, collected by individual beekeepers during the 2022 harvest (Figure 1). These areas are differentiated by differences in geographical position, climatic-environmental factors and soil composition, showing a different ecological-vegetative climate and great biodiversity. Propolis samples were randomly obtained after honey extraction by conventional scraping. After removal of debris, propolis samples were stored at −20 °C until analysis.

### 2.3. Water Content

In order to determine the free water content, it was determined following the protocol suggested by Funari et al. [13]. The sample was dried in a conventional oven at 105 °C for 2 h until constant weight was reached.

### 2.4. Ash Content

The ash content was determined following the AOAC [14] procedure, by ashing the raw propolis samples at 600 °C.

### 2.5. Wax Extraction

The wax contents were estimated according to a procedure described by Papotti et al. [15]. Three grams of frozen propolis was powdered and treated with 120 mL of petroleum ether at 40−60 °C in a Soxhlet extractor for 6 h. The extract was transferred to a previously weighed 150 mL evaporator flask and concentrated under reduced pressure at 50 °C. Then, 120 mL of 70% ethanol was added, heated under reflux until a clear solution was obtained and then cooled at 0 °C for 1 h to promote wax separation. The mixture was filtered through a previously weighed Whatman grade no. 41 filter paper. The flask and the filter were washed with 70% ethanol, dried at 110 °C for 1 h and transferred to a desiccator until constant weight. The sum of the residues remaining in the flask and on the filter, expressed as % *w*/*w*, represents the waxes.

### 2.6. Balsam Extraction and Quantification

The contents of balsams were estimated according to a procedure described by Papotti et al. [15]. The 70% ethanolic filtrate obtained during wax extraction was concentrated under reduced pressure at 60 °C. The aqueous residue was transferred to a separating funnel and 50 mL of dichlorometane was added. After shaking, the organic phase was collected and dried over 30 g of anhydrous Na_2_SO_4_ and then filtered in a previously weighed 150 mL evaporator flask. The extraction was repeated twice. The solution was evaporated to dryness under reduced pressure at 60 °C and the flask was transferred to a desiccator until constant weight. The results are expressed as % *w*/*w*.

### 2.7. Resin Extraction

The contents of resins were estimated according to a procedure described by Papotti et al. [15]. The residual propolis obtained after the extraction in the Soxhlet equipment was treated with 120 mL of a mixture of chloroform/ethanol 1:1 (*v*/*v*) in a Soxhlet extractor for 6 h. The extract was transferred to a preweighed 150 mL evaporator flask and concentrated to dryness under reduced pressure at 70 °C. The flask was dried at 110 °C for 1 h and transferred to a desiccator until constant weight. The results are expressed as % *w*/*w*.

### 2.8. Colorimetric Analysis

To determine the color indices of the propolis samples, the following were recorded: L* (lightness), a* (redness-green) and b* (yellow-blue). The colorimeter was previously calibrated using a standard white plate (L* = 94.56, a* = −0.31, b* = 4.16, C*ab = 4.18). The analysis was performed in quadruplicate.

### 2.9. Extraction of Phenolic Compounds

Pretreatment was required to determine the total phenol content and the flavonoid content. The method was suggested by Özkök et al. [16], with slight modifications. The sample was mixed with 75% ethanol/water (*v*/*v*), homogenized and sonicated in an ultrasonic bath for 5 h and finally centrifuged.

### 2.10. Total Phenols Content (TPC)

The total phenolic content was determined by a modification of the Folin–Ciocalteu method, as described by Escheriche et al. [17], with some modifications. A volume of ethanolic extracts (500 μL) were mixed with 250 μL of Folin–Ciocalteu reagent. After 3 min, 1000 μL saturated sodium carbonate solution was added to the mixture. The solution was then incubated at room temperature for 1 h and the absorbance was measured at 760 nm. The gallic acid calibration curve was used to determine the total phenolic content and the results were expressed as mg gallic acid equivalent per g propolis (0.0125 to 0.1 mg/mL). Analyses were performed in triplicate.

### 2.11. Total Flavonoid Content (TFC)

The total flavonoid content of the crude extract was determined by the aluminum chloride colorimetric method as suggested by Escheriche et al. [17], with some modifications. A volume of supernatant (50 μL) was mixed with 1500 μL of 2% aluminum chloride in methanol and 1350 μL of methanol. After 30 min of incubation in the dark at room temperature, the absorbance at 415 nm was measured. The blank test replaced the sample with distilled water and a volume of 2850 μL of methanol, placed under the same incubation conditions. Quercitin was used to calculate the standard curve (0.02 to 0.25 mg/mL) and the results were expressed as mg quercitin equivalents per g propolis. Analyses were performed in triplicate.

### 2.12. Antioxidant Power (FRAP) Method

The FRAP assay was conducted following the method described by Chaves et al. [18], with some modifications. The extract (200 μL) was mixed with 2800 μL of FRAP reagent. This reagent was previously prepared by mixing 300 mM sodium acetate buffer solution at pH 3.6, 10 mM TPZT and 20 mM FeCl3 hexahydrate, in a ratio of 10:1:1, respectively. The mixture was incubated for 30 min at 37 °C and subsequently read at 593 nm. The blank was prepared by replacing the same amount of diluted extract with methanol. The results were expressed in mM equivalents of Trolox per gram of propolis, after performing a calibration curve at known concentrations (0.01 to 0.1 mM of Trolox/mL). Analyses were performed in triplicate.

### 2.13. ABTS Free Radical Scavenging

The antioxidant activity of propolis extracts by the ABTS spectrophotometric assay was determined with the method suggested by Chaves et al. [18], with some modifications. The extract (100 μL) was mixed with 2900 μL of the ABTS^+^ dilution. The decrease in absorbance at 734 nm was measured after 30 min of incubation at room temperature. The blank was prepared with methanol only. The absorbance was read at 730 nm after 30 min. The results were expressed in mM equivalents of Trolox per gram of propolis. Analyses were performed in triplicate.

### 2.14. Statistical Analysis

Statistical analysis was performed using the general linear model (GLM) procedure of statistical analysis system SAS [19], using a monofactorial model: y_ik_ = μ + α_i_ + ε_ik_; where: μ = average mean; α_i_ = effect of geographical origin (1, …, 8); and ε_ik_ = experimental error. The Student’s *t*-test was used for all variables comparisons. Differences between means at the 95% (*p* < 0.05) confidence level were considered statistically significant. Pearson correlation coefficient (r) was used to analyze the correlations between different parameters of propolis samples.

## 3. Results

### 3.1. Physico-Chemical Composition in Propolis Samples

The physicochemical parameters of propolis samples collected in eight geographic areas of the Basilicata region are summarized in Table 1.

A statistically significant effect of geographic origin on the physicochemical parameters of propolis from eight different areas in one-way ANOVA was confirmed for all the parameters analyzed (*p* < 0.05), in agreement with Kasote et al. [5]. The moisture content of the propolis provides information on the quality of the propolis; the high water content in propolis indicates improper storage and handling conditions. The average percentage moisture content in the samples investigated was 3.57 ± 0.11, in line with what was found in Moroccan propolis by El Menyiy et al. [20] and no significant differences were found between the samples studied. The factors that influence the moisture content of propolis concern both the handling conditions and the duration of storage, which is to be considered a quality parameter, given the presence of a high content of phenols, which deteriorate easily over time. The ash content also highlighted the presence of inorganic minerals, as well as the presence of impurities present in the sample, probably linked to the natural production process that brings with it different materials, such as wood, remains of bees and small pieces of earth with a consequent increase in the level of ashes. Furthermore, determining the ash content is essential to rule out the possibility of adulteration of the propolis samples [21]. In general, the samples analyzed have an average ash content of 1.22 ± 0.04%, and the values vary from 0.68 ± 0.01 to 2.13 ± 0.21%, in agreement with what is reported in one study conducted on propolis samples from Morocco, in which values were recorded between 0.72 ± 0.02% and 5.01 ± 0.01% [22]. Furthermore, in other studies on Mexican propolis [23], the ash content ranged from 0.66% to 5.50%, with respect to the differences in the survey areas. Notably, the ash content of A-5 was significantly higher than the other propolis samples (*p* ≤ 0.05). The percentage content of the wax, resin and balsam in the propolis samples under study are shown in Table 1. An average wax content of 27.98 ± 1.68% was recorded in line with what was found by Gardini et al. [11] in Italian propolis. The highest average content among our samples was found to be 31.6 ± 2.84% for sample A-1, while propolis A-6 had the lowest value (23.66 ± 1.71%, *p* < 0.05). The differences found in the wax content could be related to the collection method rather than being influenced by the botanical and/or geographical origin of the sample. The presence of a high content of waxes, and biologically inactive components, could lead to a low percentage of pharmacologically active compounds with consequent repercussions on the commercial value of the product. The number of resins and balms in propolis is directly related to the amount of resin collected by bees during grazing. In general, an average resin content of 61.32% was recorded, while, for the conditioner, it was 6.13%. Some variability was observed for the resin and balsam content between the samples, ranging from 48 to 71% and 4.36 to 9.61%, respectively. In the comparison between the samples, A-6 and A-8 had a resin content significantly higher than 65% (*p* < 0.05). Regarding the balsam content, sample A-5 showed the highest content (9.63%; *p* < 0.05), while the lowest content was found in A-1 and A-8 propolis (*p* < 0.05). The resins and balms contain bioactive plant metabolites that perform numerous biological activities, contributing to the defense of the hive. Papotti et al. [15] highlighted the relationship between the chemical composition and health of bee colonies, in particular the level of resins and balsams contained in propolis. Drescher et al. [3] found, in fact, that the resin content was significantly lower in the colonies more resistant to Varroa and, therefore, the bees resistant to the parasitic mite reserved few resources for resin collection, compared to the bees coming from particularly sensitive colonies. Color is a determining physical-chemical parameter in the choice of the product. The colorimetric characteristics of propolis samples from different geographical areas are shown in Figure 2.

The values, measured with the CIE L*a*b* method, showed high and consistent variability between the propolis samples from different production areas, resulting in being statistically significant (*p* < 0.05). In general, the samples were dark, with L* values ranging from 42.3 to 50.15; in particular, A-3 and A-1 recorded the lowest value of L* (respectively, 42.53 and 43.96), while the brightest were propolis A-4 (L* = 50.15). The parameters a* (red-green) and b* (yellow-blue) of the propolis can be interpreted as a reliable index of the richness in pigments of botanical origin. These values ranged from 7.43 (A-4) to 11.1 (A-1) for parameter a* and from 16.92 (A-3) to 19.45 (A-5) for parameter b*. These values were in line with Portuguese propolis investigated by Gomes et al. [24].

### 3.2. Content in Total Phenols and Flavonoids

In this study, the content of the total phenols and flavonoids was determined as represented in Figure 3. da Silva et al. [25] suggested that the quality of propolis is based on the content of flavonoid and phenolic compounds since they represent the major bioactive components of propolis, found mainly in resins and balms. The total phenolic compounds ranged between 221 and 461 mg GAE/g propolis (Figure 3).

Although the samples were collected from areas not far from each other, there were recorded significant differences in their total phenolic content (*p* < 0.05). Propolis A-8 presented the significantly higher total phenol content (442.26 mg GAE/g; *p* < 0.01) while A-1 and A-3 showed the lowest values (222.44 and 234.82 mg GAE/g, respectively). The TPC in propolis extracts from various parts of the world has been extensively studied and a wide range of values can be found in the literature [16]. Turkish propolis reported a range of values of 115–210 mgGAE/g; values from 151 to 329 mgGAE/g were found in Portuguese propolis by Gomes et al. [24]; in Chile and Spain, the recorded values ranged from 200 to 300 mgGAE/g [26], while, in Greek and Cypriot propolis, the range included values from 80 to 338 mgGAE/g [27]. In the analyzed samples, the flavonoid content ranged from 64.35 mgQE/g to 115.62 mgQE/g, and the highest TFC was found for propolis A-8 and A-5 (115.62 and 111.02 mgQE/g, respectively). The lowest TFC was determined for propolis A-1 and A-4 (64.35 and 64.42 mgQE/g, respectively). From our results, it has been demonstrated that the variation is rather limited compared to the data reported in the literature. By way of example, Chinese (8.3–162 mgQE/g) and Australian (0.2–144.8 mgQE/g) propolis observed much wider ranges in total flavonoid content. Furthermore, Kumazawa et al. [28] studied the content of the total flavonoids in propolis from various regions of the world, recording a relatively large variability (2.5–176 mgQE/g). In the literature, very high variability between the studied samples of propolis has been confirmed and the observed differences may derive from various factors: soil composition, temperature, humidity and altitude, which influence the physiological state of the plant and, therefore, on the phenolic biosynthesis.

### 3.3. Antioxidant Activity in Propolis Samples: ABTS and FRAP

In Figure 4, the antioxidant activities of ABTS and FRAP have been reported. It is well known that propolis exhibits strong antioxidant activity [16] and, in this present work, the ABTS and FRAP assays were chosen for the antioxidant evaluation of the propolis. The ABTS assay highlights the activity of hydrophilic and lipophilic antioxidants, while the FRAP assay uses antioxidants as reducing agents in a redox-linked colorimetric method, employing an easily reduced oxidant system present in stoichiometric excess [29]. Based on antioxidant tests, propolis from different geographic areas has been observed to exhibit varying degrees of antioxidant capacity. The mean values were 5.41 and 1.48 mMTE/g for the ABTS and FRAP assays, respectively. Significant differences have been recorded between the propolis from different locations, suggesting that they have different antioxidant potentials.

As shown in Figure 4, the values ranged from 4.97 (A-5) to 5.66 mMTE/g (A-3) in the ABTS assay and from 1.2 (A-3) to 1.76 mMTE/g (A-8) in the FRAP assay.

The results obtained by means of the ABTS assay showed a lower antioxidant activity in propolis A-2 and A-5 compared to the propolis of the other areas considered (5.22 and 4.97 mM TE/g, respectively), while the propolis A-3 showed the highest radical scavenging activity (5.66 mM TE/g; *p* < 0.05). The trend of the antioxidant activity, by the FRAP assay, did not confirm the results obtained by the ABTS assay; in particular, the propolis A-3 samples showed the lowest FRAP values (1.2 mM TE/g; *p* < 0.05) than the others, while the maximum activity was observed for propolis sample A-8 (1.76 mMTE/g; *p* < 0.01). Our results showed higher ABTS values than those reported by Martín et al. [30] in Spanish propolis, where they detected values of 1.823 mmol TE/g. The results obtained in this work confirmed and demonstrated that the variations recorded in the antioxidant activity are influenced by the different collection locations, which differ in geographical and climatic factors and in the different composition of the soil. These factors greatly influence the content of biologically active compounds in propolis, which can act synergistically and increase the antioxidant action. A significant and positive linear correlation was observed between FRAP and TPC (r = 0.488; *p* < 0.01) and FRAP and TFC (r = 0.753; *p* < 0.01) in agreement with Kasote et al. [31], who observed how propolis characterized by a high content of phenolic compounds has shown a strong antioxidant activity. In contrast, low and negative correlation coefficients were observed between TPC (r = −0.033) and TFC (r = −0.199) with the ABTS assay; this means that the antioxidant activity of the propolis sample could be due to other non-phenolic components present. Indeed, propolis is characterized by an abundant presence of phytochemicals, including essential oils, minerals and vitamins A, B, C and E, as suggested by Sahlan et al. [32], in which they defined the important and specific role of these components in biological activities. In agreement with our data, propolis from several countries such as Argentina [30], Greece and Cyprus [27], Japan [33] and Poland [33] showed a high correlation between TPC and TFC and the scavenging activity of free radicals. Instead, a negative or absent correlation between them was observed both in the propolis of Morocco and in the propolis of Brazil [22] but also in that of Greece [27]. Our data revealed that propolis with a high content of resin, phenolics and flavonoids had the highest antioxidant activity and that a high amount of flavonoids and phenols was found in samples with a high resin content and low wax content, in line with what was found in Moroccan propolis [20]. However, the number of polyphenols is strongly influenced by the climatic conditions of each collection area, which explains these large differences between the studied samples. Similar data were observed in Brazilian propolis by da Silva et al. [25], in which the relationship between climatic conditions, metabolite profile and antioxidant activity emerged.

## 4. Discussion

This study summarizes for the first time the state of knowledge on the characteristics of propolis produced in Basilicata and the factors to be considered to characterize the quality of honey.

Overall, it can be concluded that Lucanian propolis has a low wax content and a high content of resins, balms and antioxidant compounds with a marked antioxidant capacity. The chemical, physical, or biological properties of “Lucana” propolis varied considerably between the different propolis according to the geographical location and the diversity of the flora surrounding the apiary that the bees use as a source of resin. Although further work is needed to investigate and define a complete picture of the propolis of the Basilicata region with regard to their chemical composition and therapeutic values, the results of this study provide the basis for the production and commercialization of propolis not only in the food industry but also in the pharmaceutical and cosmetic fields.

## Figures and Tables

**Figure 1 foods-12-03481-f001:**
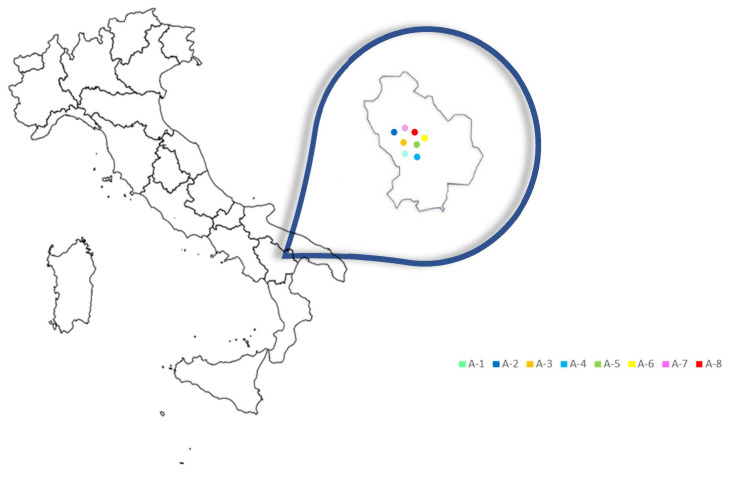
Distribution of propolis sampling areas in the 8 areas in the Basilicata region. Each area is 30 to 40 km apart from each other.

**Figure 2 foods-12-03481-f002:**
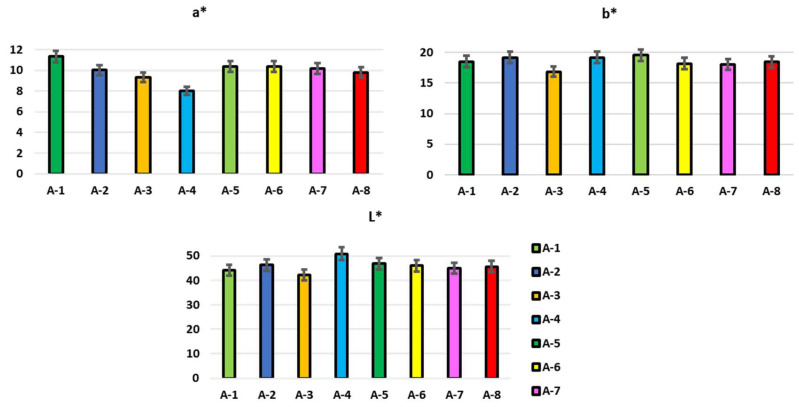
Colorimetric parameters of propolis samples from different geographical areas (*p* < 0.05).

**Figure 3 foods-12-03481-f003:**
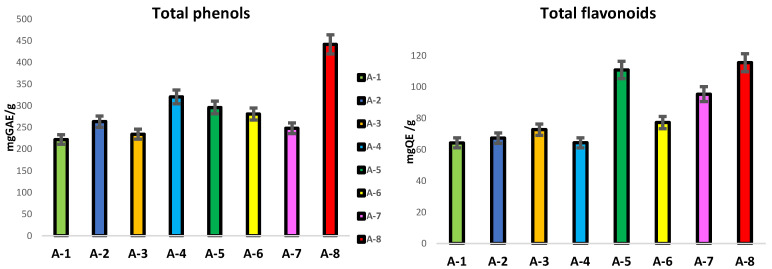
Total phenols (mgGAE/g) and total flavonoids (mgQE/g) of propolis samples from different geographical areas (*p* < 0.05).

**Figure 4 foods-12-03481-f004:**
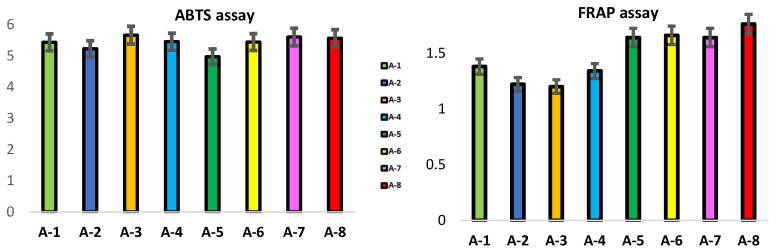
Antioxidant assays (ABTS and FRAP, mMTE/g) of propolis from different geographic areas.

**Table 1 foods-12-03481-t001:** Physico-chemical composition (%) in fresh propolis differently by collection area.

	Dry Matter		Ash		Wax		Resin		Balsam	
	µ	^1^ SD	µ	^1^ SD	µ	^1^ SD	µ	^1^ SD	µ	^1^ SD
A-1	3.72 ^ab^	0.31	1.75 ^a^	0.09	31.55 ^a^	1.82	48.67 ^a^	1.67	4.33 ^a^	0.2
A-2	3.23 ^a^	0.30	1.01 ^b^	0.07	30.4 ^ab^	1.47	56.49 ^b^	2.25	6.39 ^b^	0.23
A-3	3.39 ^ab^	0.18	0.89 ^c^	0.05	29.08 ^bc^	1.01	59.81 ^c^	1.23	6.4 ^b^	0.26
A-4	3.79 ^b^	0.28	1.84 ^d^	0.03	27.88 ^c^	1.97	64.3 ^d^	1.36	7.82 ^c^	0.51
A-5	3.78 ^ab^	0.32	0.77 ^e^	0.03	27.55 ^c^	1.61	63.4 ^d^	1.46	4.42 ^a^	0.42
A-6	3.31 ^a^	0.28	2.06 ^f^	0.15	24.47 ^d^	1.57	71.78 ^e^	1.51	9.63 ^d^	0.73
A-7	3.53 ^ab^	0.25	0.7 ^g^	0.03	30.9 ^a^	1.45	58.38 ^bc^	1.75	5.41 ^e^	0.52
A-8	3.82 ^ab^	0.27	0.89 ^c^	0.05	24.4 ^d^	1.15	68.39 ^f^	1.38	4.54 ^a^	0.44

^1^ SD standard deviation; ^a,b,c,d,e,f,g^ means within a column with different superscripts differ (*p* < 0.01).

## Data Availability

The data that support the findings of this study are available from the corresponding author upon reasonable request.

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
