# Peer review of "Chemical and Functional Characterization of Propolis Collected from Different Areas of South Italy"

_foods, 2023, doi:10.3390/foods12183481_

Round 1

Reviewer 1 Report

The authors investigated physical and chemical properties of propolis collected from different regions of South Italy. The significance of this study lies in the additional information it provides about propolis from various geographical regions, especially South Italy. The methodologies and results are clear as they were standardized methodologies. However, for clarity purposes, please amend figures 2,3,4 into 2-D graphs with error bars where appropriate.  

Overall, the manuscript was written in good English, except for some minor errors. For example, numbers should not be used at the beginning of a sentence. 

Author Response

Authors' response

Many thanks to the reviewers who took the time to review this manuscript. Below are the detailed responses and corresponding revisions/corrections highlighted/tracking changes in the resubmitted files

Reviewer 1:

- Comments and Suggestions for Authors

The authors investigated physical and chemical properties of propolis collected from different regions of South Italy. The significance of this study lies in the additional information it provides about propolis from various geographical regions, especially South Italy. The methodologies and results are clear as they were standardized methodologies. However, for clarity purposes, please amend figures 2,3,4 into 2-D graphs with error bars where appropriate.  

Author's answer:

I thank reviewer for appreciating the research work as a whole. The graphs have been modified according to your suggestion.

- Comments on the Quality of English Language

Overall, the manuscript was written in good English, except for some minor errors. For example, numbers should not be used at the beginning of a sentence. 

Author's answer:

Thank you again for the suggestion. I will make the suggestions as requested. Particularly, in the “Materials and Methods” section.

Reviewer 2 Report

Grassi and colleagues characterised sixteen propolis samples collected in the Basilicata region of Italy. Finally, the results of eight regions were presented. They analysed the essential components, phenols, flavonoids, and antioxidant capacity. The authors compared the values of their results with other propolis samples analysed and published worldwide.

The manuscript is clear, relevant to the field, and well-presented in a structured manner. It comprises four main sections: Introduction, Materials and Methods, Results, and Discussion with references.

The references cited are mostly recent (published in the last five years) and relevant publications. No self-citation.

The manuscript meets scientific standards, and the experimental design is adequate.

Methods are adequately described, which gives a chance of reproducibility, but the nature of the samples partly precludes reproducibility, as they were collected at a specific place and period.

One table and four figures are provided. They are clear and informative.

The statistical methods chosen are appropriate and well-used.

The conclusions are broadly consistent with the evidence and arguments presented.

No ethical statement was required based on the study design. The authors provide supporting data on request.

Overall, the manuscript is well written. It describes the parameters of local propolis samples, which can thus be compared with propolis samples analysed elsewhere in the world.

Comments:

1. Lucanian propolis was highlighted but needed to be better defined. Which one is it, A-6? Where is it from in the Basilica region? What is the reason for its specific properties?

2. The references should be revised, as there are some publications among the references that are not mentioned in the main text: Afrouzan et al., Alanazi et al., Nakamura et al., Özkök et al., Pahlavani et al. and Santos et al.

Author Response

Authors' response

Many thanks to the reviewers who took the time to review this manuscript. Below are the detailed responses and corresponding revisions/corrections highlighted/tracking changes in the resubmitted files

Reviewer 2:

- Comments and Suggestions for Authors

  1. Lucanian propolis was highlighted but needed to be better defined. Which one is it, A-6? Where is it from in the Basilica region? What is the reason for its specific properties?

Author's answer:

I thank the reviewer for the suggestion. However, I believe that propolis has been extensively described in the "Introduction" section trying to delve deeper into the intrinsic and extinct characteristics of the product. Furthermore, the collection points in the Basilicata region fall within a radius of 30 to 40 km as described in the section "2.2. Propolis samples" and illustrated in Figure 1.

- Comments and Suggestions for Authors

  1. The references should be revised, as there are some publications among the references that are not mentioned in the main text: Afrouzan et al., Alanazi et al., Nakamura et al., Özkök et al., Pahlavani et al. and Santos et al.ù

Author's answer:

Thanks to the reviewer for the suggestion. The references have been correctly modified in order to have coherence between citations in the text and in the bibliography, as it should be. (See References and text).

Reviewer 3 Report

Reviewer

Chemical and Functional Characterization of Propolis Collected from Different Areas of South Italy

The aim of the presented work is interesting and very important for the local market in order to ensure the quality of domestic propolis

Some suggestions for Authors:

Line 2: delete comma at the end of the title

Line 4: who is corresponding author, mark the name with *

Line 30: In the text, reference numbers should be placed in square brackets [ ] and placed before the punctuation; for example [1], [1–3] or [1,3] – please correct in whole paper

Line 49: delete first name of Kasote

Line 108: insert space after 2023

Line 112: insert comma at the end of the sentence

Line 192: this is Results and discussion section, not only results

Line 193: delete comma at the end of the sentence – also in the other subsections

Line 211: uniform in the whole paper - 0.72 ± 0.02% or 0.72±0.02%, also (P ≤ 0.05) or (P≤0.05). please check

Line 241: please edit the table according to the instructions for authors in the Word template (https://www.mdpi.com/journal/foods/instructions)

Line 345: this is Conclusion section, not the discussion part

Line 371: this reference is not in text

Line 373: also

Line 430: also

Line 443: also

Line 446: also

Author Response

Authors' response

Many thanks to the reviewers who took the time to review this manuscript. Below are the detailed responses and corresponding revisions/corrections highlighted/tracking changes in the resubmitted files

Reviewer 3:

- Comments and Suggestions for Authors

Line 2: delete comma at the end of the title

Author's answer:

Thanks to the reviewer for the suggestion but I don’t see comma at the end of the title: Chemical and functional characterization of propolis collected from different areas of south Italy.

Line 4: who is corresponding author, mark the name with *

Author's answer:

Thanks to the reviewer for the suggestion. The corresponding author is Giulia Grassi and * was put.

Line 30: In the text, reference numbers should be placed in square brackets [ ] and placed before the punctuation; for example [1], [1–3] or [1,3] – please correct in whole paper

Author's answer:

Thanks to the reviewer. The references have been correctly arranged as requested both in the text and in the “References” section.

Line 49: Clear Kasote's name

Author's answer:

I thank the editor. The reference has been edited with the correct formatting

Line 108: Insert space after 2023

Author's answer:

I thank the editor. Formatting has been changed.

Line 112: Insert comma at the end of the sentence

Author's answer:

I thank the editor. The formatting has been changed but replaced by a period to avoid overthinking.

Line 192: This is the Results and discussions section, not just results

Author's answer:

I thank the editor. The section has been edited as required. In this section the Discussion item was eliminated because a point was subsequently dedicated to the Discussion chapter.

Line 193: Drop the comma at the end of the sentence – also in the other subsections

Author's answer:

As suggested, the point has been deleted. Thank you for your suggestion
